# MoleculeCLA: Rethinking Molecular Benchmark via Computational Ligand-Target Binding Analysis

## Abstract

Molecular representation learning is pivotal for various molecular property prediction tasks related to drug discovery. Robust and accurate benchmarks are essential for refining and validating current methods. Existing molecular property benchmarks derived from wet experiments, however, face limitations such as data volume constraints, unbalanced label distribution, and noisy labels. To address these issues, we construct a large-scale and precise molecular representation dataset of approximately 140,000 small molecules, meticulously designed to capture an extensive array of chemical, physical, and biological properties, derived through a robust computational ligand-target binding analysis pipeline. We conduct extensive experiments on various deep learning models, demonstrating that our dataset offers significant physicochemical interpretability to guide model development and design. Notably, the dataset's properties are linked to binding affinity metrics, providing additional insights into model performance in drug-target interaction tasks. We believe this dataset will serve as a more accurate and reliable benchmark for molecular representation learning, thereby expediting progress in the field of artificial intelligence-driven drug discovery.

## 1 Introduction

Molecular Representation Learning (MRL) is crucial in leveraging artificial intelligence for drug discovery applications. These applications span a range of areas, including molecular property prediction Hu et al. (2019); Hou et al. (2022); Xia et al. (2022); Zhou et al. (2023); Feng et al. (2023c), molecular dynamics simulation Zaidi et al. (2022); Feng et al. (2023b), chemical reaction prediction Wang et al. (2021); Tang et al. (2024), drug-target interactions Feng et al. (2023a), and high-throughput drug virtual screening Gao et al. (2024). The MRL approach uses deep learning models to encode molecules into meaningful latent representations, effectively capturing and preserving their molecular properties. The success of models heavily depends on the quality of evaluation datasets, which reveal limitations and guide improvements in model design. From the aspect of application, a more appropriate model can promote specific application scenarios.

There are now several datasets available for evaluating molecular representation learning models Wu et al. (2018); Ramakrishnan et al. (2014); Chmiela et al. (2017); Davis et al. (2011); Tang et al. (2014). Among them, MoleculeNet Wu et al. (2018) stands out as the most frequently utilized benchmark for evaluating molecular representation models, especially within the domain of molecular property prediction. It consists of a comprehensive range of properties sourced from various public datasets, making it an acknowledged standard for assessing the efficacy of diverse molecular machine learning methods. However, despite its popularity, MoleculeNet presents several challenges, including: **1) Data Volume Constraints.** A considerable number of the properties documented in MoleculeNet are derived from costly and intricate wet experiments, resulting in a limited dataset size. As illustrated in Figure 1a, more than half of the tasks in the MoleculeNet dataset comprise fewer than 10,000 data points. In such scenarios, models frequently encounter overfitting issues, which may worsen under stricter split settings, such as scaffold split. **2) Unbalanced Label Distribution.** Many of MoleculeNet's datasets exhibit severe label imbalances that can distort performance metrics. Figure 1b illustrates the proportions of samples whose label equals 1 in all multi-label binary classification tasks within MoleculeNet. It is evident that the majority of proportions tend to be closer to 0 or 1, indicating

the prevalence of the unbalanced issue of MoleculeNet. **3) Label Noise.** The dependency on wet experiments introduces a degree of uncertainty in the dataset labels, some of which may be imprecise due to the inherent limitations of experimental methods. This factor compromises the reliability of the data. As pointed out by Walters (2023), numerous flaws exist within the MoleculeNet, including invalid structures, undefined stereocenters, and conflicting labels caused by data curation errors.**4) Inconsistency.** MoleculeNet compiles its data from several public databases, assigning different molecular sets to each property task. This aggregation process not only leads to inconsistencies but also allows batch effects to manifest. In summary, the various issues outlined make the results of MoleculeNet unstable and susceptible to influence from factors such as varying hyperparameters and random seeds. Consequently, many existing methods Zhou et al. (2023); Feng et al. (2023c); Yu et al. (2023) resort to hyperparameter search techniques in pursuit of improved performance. However, the dominance of hyperparameters over the method itself compromises the reliability of the benchmark and poses challenges in accurately profiling methods.

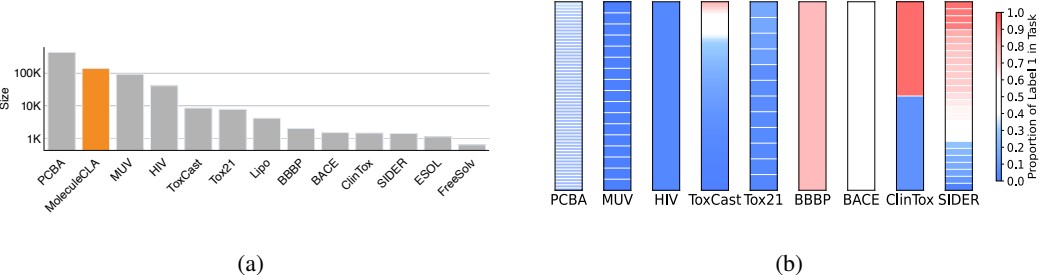

(a)             (b)

Figure 1: The statistical analysis of data numbers and label distribution about tasks in MoleculeNet. (**a**) indicates that the majority of task datasets consist of fewer than 10,000 entries. (**b**) illustrates the label distribution across all subtasks within each classification task. It is obvious that the proportions of samples with a label value of 1 show a bias towards either extreme, indicating a significant imbalance issue in MoleculeNet's label distribution.

To address these issues, we propose MoleculeCLA: a large-scale benchmark for **molecular** property prediction via **c**omputational **l**igand-target binding **a**nalysis(Figure 2). Ligand-target binding, which is an important task in drug discovery, is a complicated process influenced by various molecular properties Decherchi & Cavalli (2020). Consequently, docking tools are designed to incorporate many different meaningful components to fit the final docking score. From these components, we meticulously select diverse items relevant to physical, chemical, and biological molecular properties, outlined in detail in Table 1. Notably, MolecueCLA is derived from a computational approach that does not rely on wet experiments. This method not only enhances data accessibility but also scales conveniently to accommodate large amounts of data.

Building on the above content, in this work, we construct a large-scale and precise dataset involving 10 representative protein targets spanning a wide range of drug functionalities. We use the docking software Glide Friesner et al. (2004); Halgren et al. (2004) to obtain 9 properties for a total of 140,697 molecules, covering chemical, physical, and biological properties. We evaluate various MRL methods, including traditional descriptor-based and deep-learning approaches. The performance results are more stable and provide better explanations about the methods themselves, indicating that our dataset is more robust and reliable. Additionally, these properties have a strong correlation with binding affinity, enabling us to further use this dataset to select appropriate models for drug-target interaction tasks.

## 2 DATASET

We explore a new paradigm for cultivating benchmarks through computational approaches for the evaluation of MRL methods, avoiding expensive and noisy wet experiments while easily scaling to large datasets. Our focus is on the ligand-protein binding process, which is closely related to drug discovery. We extract chemical, physical, and biological properties of molecules from ligand-target binding analyses using Glide. Each molecule binds to multiple protein targets in the same quantity.

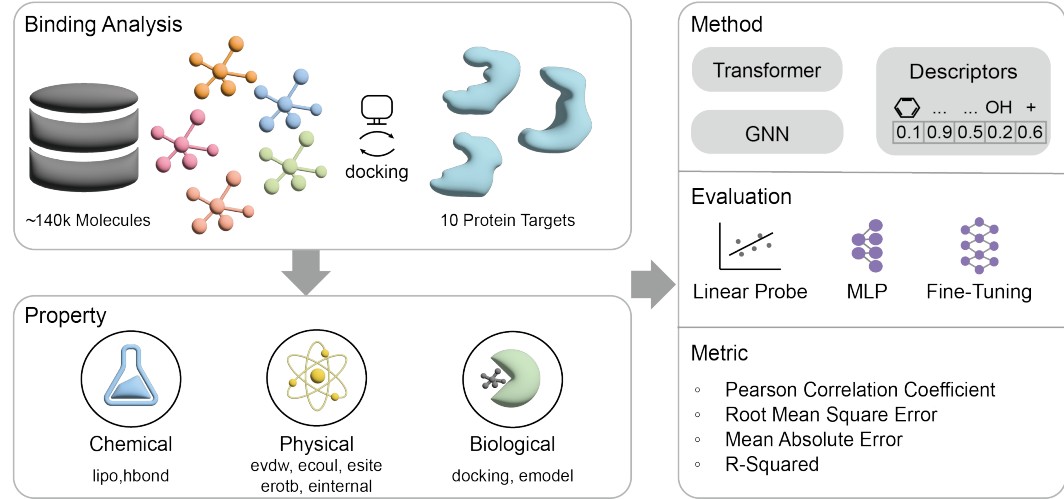

Figure 2: The overview of MoleculeCLA: diverse categories of molecular properties are derived from the computation binding analysis. We assess methods like deep learning models and descriptors through linear prob, MLP, and fine-tuning testing protocols. Results are presented via multiple regression task metrics.

Tasks are organized and split by binding targets. In each task, we evaluate the model by regressing different molecular properties relevant to binding to the same target, which can be treated as a constant environment.

Table 1: **Glide Properties for Molecular Property Benchmarking.** Nine Glide-calculated molecular properties are summarized and categorized into chemical, physical, and biological aspects. Each property is described with its abbreviation, along with a brief explanation and the underlying molecular characteristics it reflects.

| Aspect | Glide Property (Abbreviation) | Description | Molecular Characteristics |
|---|---|---|---|
| Chemical | glide_lipo (lipo)
glide_hbond (hbond) | Hydrophobicity
Hydrogen bond formation propensity | Atom type, number
Atom type, number |
| Physical | glide_evdw (evdw)
glide_ecoul (ecoul)
glide_esite (esite)
glide_erotb (erotb)
glide_einternal (einternal) | Van der Waals energy
Coulomb energy
Polar thermodynamic contribution
Rotatable bond constraint energy
Internal torsional energy | Size and polarizability
Ionic state
Polarity
Rotational flexibility
Rotational flexibility |
| Biological | docking_score (docking score)
glide_emodel (emodel) | Docking score
Model energy | Binding affinity
Binding affinity |

## 2.1 Data Collection and Processing

We select approximately 14,000 highly diverse drug-like molecules from popular commercially available libraries to ensure a broad representation of chemical space. These small molecules undergo Glide typical computational ligand-target binding analysis protocols, including ligand preparation, protein preparation, grid generation, and molecular docking. The docking results are meticulously inspected and selected.

We carefully select 9 different properties generated by Glide, categorized into three groups: chemical, physical, and biological. Details about these properties are listed in Table 1. In the chemical property collection, *glide_lipo* and *glide_hbond* are included, describing hydrophobicity and hydrogen bond formation, respectively. These properties are directly linked to the chemical composition of the molecule itself (i.e., atom type and number). The physical property collection includes *glide_evdw*, *glide_ecoul*, *glide_erotb*, *glide_esite*, and *glide_einternal*, all of which are computationally obtained energy values (the 'e' in the property names stands for 'energy'). These properties are mainly deter-

mined by the physicochemical characteristics of the molecules. The biological property collection includes *docking_score*, and *glide_emodel*, which represent the Glide-predicted binding affinity. These properties are more correlated with the ligand-target binding process.

To comprehensively capture the properties exhibited by small molecules when interacting with various targets, we chose 10 representative targets from multiple categories. These targets include both human and viral proteins, covering a diverse range of biological functions and structural characteristics. This selection ensures broad coverage of potential interactions, allowing Glide property calculations to capture the behavior of small molecules across different target types, thereby revealing multiple binding characteristic profiles for each molecule. The protein targets within each category are carefully chosen, many of which are well-investigated drug targets. For instance, the 3C-like protease (3CL) of SARS-CoV-2 is a critical target for the treatment of mild-to-moderate COVID-19 Consortium et al. (2020). Detailed information on all protein targets can be found in Table 2.

Table 2: **Summary of Protein Targets.** The *Category* column indicates the classification of each protein target, while the *Name* column specifies the name of each protein target. The *Resolution* column denotes the resolution of the protein structure in ångströms (Å), and the *PDB ID* column lists the Protein Data Bank identification numbers for each protein.

| Category | Name | Resolution | PDB_ID |
|---|---|---|---|
| Kinase | ABL1 | 1.74 | 3K5V |
| G-Protein Coupled Receptor | ADRB2 | 2.70 | 5X7D |
| Ion Channel | GluA2 | 2.72 | 8SS9 |
| Nuclear Receptor | PPARG | 1.95 | 3ET3 |
| Cytochrome | CYT2C9 | 2.00 | 1R9O |
| Epigenetic | HDAC2 | 1.26 | 7KBG |
| Viral | 3CL | 1.18 | 7GEF |
| | HIVINT | 1.80 | 3NF7 |
| Others | KRAS | 1.01 | 8ONV |
| | PDE5 | 1.30 | 1TBF |

## 2.2 DATASET SPLIT

Generalization is a crucial aspect of model evaluation, so we use scaffold splitting to minimize data leakage and ensure robust testing. Scaffold splitting ensures that structurally diverse molecules populate the training, validation, and test sets, exposing the model to a wide range of molecular scaffolds and enhancing its generalization to new, unseen molecules, thereby mimicking real-world situations. Specifically, our dataset is divided using scaffold splitting, resulting in a training set of 112,557 molecules, a validation set of 14,070 molecules, and a test set of 14,070 molecules. This division is used in all subsequent experiments.

## 2.3 DATASET ANALYSIS

**Molecular Chemical Space Coverage** To demonstrate the diverse chemical space of MoleculeCLA, we extract the Extended-Connectivity Fingerprints (ECFP) Rogers & Hahn (2010) of molecules in MoleculeCLA along with other binding-related molecular benchmarks: PCBA Wang et al. (2012) from MoleculeNet, MoleculeACE, KIBA, Davis, and LBA. We then use the t-SNE Van der Maaten & Hinton (2008) algorithm to visualize molecules from these different datasets and compare the chemical spaces they cover. As shown in Figure 3a, the sample points from MoleculeCLA and PCBA cover the most area of the entire sample space. Given that MoleculeCLA contains only about one-third the number of molecules in PCBA, this demonstrates that MoleculeCLA encompasses a rich chemical space and has the potential to cover an even larger space when scaled to an equal number of molecules.

**Label Distribution**: We illustrate the distribution of label values in our 9 regression tasks in Figure 3c. Most task labels appear smooth, except for *hond* and *esite*, as hydrogen bond and polar interactions are rare and often zero. As demonstrated in the experiments of Section 3, these two tasks are the most

challenging for baseline models due to their nearly discrete distributions. Furthermore, we provide the mean and standard deviation values of samples in the train, validation, and test sets for different tasks under the scaffold split in Table 7.

**Task Diversity** Although the 9 molecular properties in MoleculeCLA originate from the same docking software, they are diverse and represent different aspects of molecular properties. To demonstrate this, we calculate the Pearson correlation of labels between each pair of tasks and illustrate the correlation matrix in Figure 3b. It is evident that the labels of each task exhibit weak correlations, indicating that our tasks are diverse and can profile methods from different aspects.

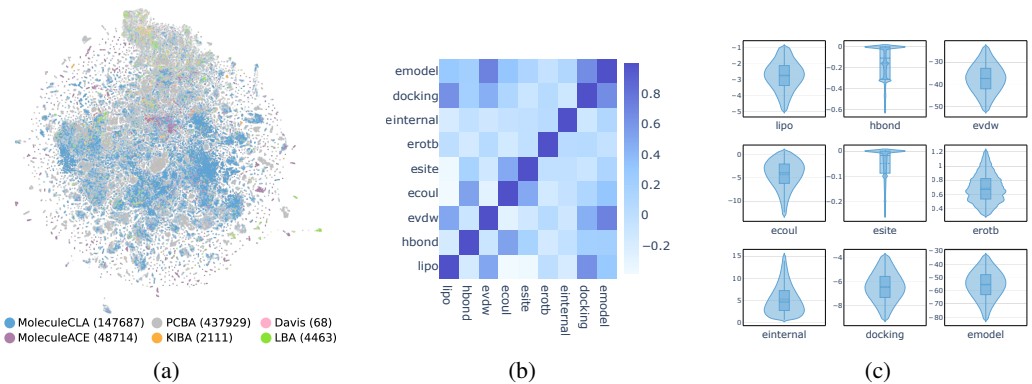

(a)  (b)  (c)

Figure 3: Data analysis of MoleculeCLA: (**a**) The t-SNE visualization of fingerprint clustering across various datasets, including MoleculeCLA, PCBA, MoleculeACE, KIBA, Davis, and LBA, reveals that despite containing approximately one-third the number of samples, MoleculeCLA demonstrates a chemical space comparable to PCBA. (**b**) The Pearson correlation matrix among tasks within MoleculeCLA showcases the diversity of different properties. (**c**) Examining the label value distribution across all nine tasks, most tasks exhibit smooth distributions, with the exception of *esite* and *hbond*.

## 3 EXPERIMENTS AND RESULTS

Our dataset comprises chemical, physical, and biological properties, and to comprehensively evaluate baseline models, we assess them based on their latent representation and model transfer ability. For latent representation, we employ the linear probe method. This method is widely utilized in deep learning model evaluation and provides a sole assessment of the quality of the representation and insights into what abstract information the model has captured Akhondzadeh et al. (2023); Radford et al. (2021); Liang et al. (2022). In terms of transfer ability, we commonly load the parameter values from the pre-trained model to process the input, then update all parameters including the molecular encoder to predict the task label as a standard fine-tuning paradigm. Additionally, we fine-tune models to assess their performance in the drug-target interaction task. This evaluation can reflect whether our dataset can be used to choose the appropriate model for the drug-target interaction task.

### 3.1 BASELINES AND EVALUATION METRICS

Molecular representation learning encompasses various model architectures, primarily classified into Graph Neural Network (GNN)-based models and Transformer-based models, which handle different formats of molecular representation such as SMILES strings, 2D graph structures, or 3D coordinates. For robust and comprehensive evaluation, we have chosen nine representative deep learning models known for their superior performance as our baseline. A brief description of these models is provided in Table 3, more details can be found in the Appendix 5.4

Our dataset is organized into 9 regression tasks for each of the 10 targets, as the properties consist of continuous values. We use the Pearson correlation coefficient, Root Mean Square Error (RMSE), Mean Absolute Error (MAE), and R-squared (R2) as our regression evaluation metrics. The Pearson correlation coefficient helps in understanding the linear relationship between variables, RMSE and MAE provide insights into the magnitude of prediction errors, and R2 offers a measure of how well

the model captures the variability in the data. Together, these metrics ensure a robust evaluation of model performance.

Table 3: **Summary of Deep Learning Models.** The *Input* column specifies the molecular representation format. *Architecture* identifies the model's backbone network structure, such as a Graph Neural Network (GNN) and Transformer. *Strategy* notes the pre-training approach, such as Masked Component Modeling (MCM), and Contrastive Learning(CL), Denoising. *Dim* lists the dimension of the model's latent molecular representation.

| Model | Input | Architecture | Strategy | Dim |
|---|---|---|---|---|
| AttrMasking Hu et al. (2019) | Graph | GNN | MCM | 300 |
| GraphMAE Hou et al. (2022) | Graph | GNN | MCM | 300 |
| Mole-BERTXia et al. (2022) | Graph | GNN | MCM | 300 |
| Uni-Mol Zhou et al. (2023) | 3D Coordinates | Transformer | Denoising, MCM | 512 |
| Uni-Mol+ Lu et al. (2023) | 3D Coordinates | Transformer | Denoising | 768 |
| 3D Denoising Zaidi et al. (2022) | 3D Coordinates | GNN | Denoising | 256 |
| Frad Feng et al. (2023b) | 3D Coordinates | GNN | Denoising | 256 |
| SliDe Ni et al. (2023) | 3D Coordinates | GNN | Denoising | 256 |
| UniMAP Feng et al. (2023c) | SMILES, Graph | Transformer | MCM, CL | 768 |

## 3.2 Comparative Analysis of Model Latent Representation Using Linear Probe

### 3.2.1 Experimental Setup

In linear probe experiments, we evaluate the latent representations of various baseline models and molecular descriptors. Molecular descriptors are mathematical representations of a molecule's properties. Unlike the abstract latent representation learned by the model, the dimensions in the descriptor vector correspond directly to different physical and chemical information of the molecules, e.g., topological polar surface area and the number of all atoms. We chose the widely used molecular descriptor calculation software Mordred Moriwaki et al. (2018) to obtain the 2D and 3D descriptors. After generating the descriptors, we drop columns containing empty values. Following this process, the 2D descriptor dimension is 904 and the 3D descriptor dimension is 51. Then, molecular descriptors and latent representations are used as input to train a linear regression model implemented with the Scikit-learn package Pedregosa et al. (2011).

### 3.2.2 Results & Analysis

The average Pearson correlation coefficients for 10 protein targets using the linear probe method are presented in Table 4. These results provide a deeper understanding of the effectiveness of various molecular representation learning models. Below, we explore specific findings that reveal how these models encode molecular information, offering valuable insights into model development and design. We also provide the RMSE, MAE, R2 results in Appendix 5.6.

**Descriptors Provide More Direct Molecular Information** Descriptors$_{3D}$ have only 51 features, but its overall performance is still better than that of the AttrMasking and GraphMAE. Descriptors$_{2D}$ and Descriptors$_{2D\&3D}$ have similar results across all tasks and outperform the limited-feature Descriptors$_{3D}$. Therefore, in the following discussion, "descriptors" will refer to Descriptors$_{2D}$. As observed in Table 4, descriptors consistently achieve the highest Pearson correlation coefficients across all tasks. This result may be due to the inclusion of certain features in the descriptors, such as acidic group count, number of hydrogen atoms, and hybridization ratio, which are directly correlated with properties like lipophilicity, hydrogen bonding, electrostatic interaction, and others. However,

when we use baseline model latent representations and molecular descriptors as input to train a multi-layer perceptron model for each task, as detailed in the Appendix 5.5, we find that some model latent representations outperform the descriptors. This indicates that while descriptors provide more direct molecular information, model latent representations may contain more abstract semantics, leading to better results when they are used with more expressive downstream models.

**Appropriate Mask Strategy Potentially Enhances Graph-based Models**   The analysis of results from three GNN models (AttrMasking, GraphMAE, and Mole-BERT) that utilize graph data as input but employ different masking strategies suggests that Mole-BERT potentially performs better than the others. AttrMasking randomly masks some atoms and then predicts the masked atom types. GraphMAE further employs a re-mask decoding strategy to reconstruct atom features. In contrast, Mole-BERT identifies a challenge with masking atom types due to the limited and unbalanced nature of atom sets. Instead, it employs a context-aware tokenizer, encoding atoms into chemically relevant discrete values for masking, thereby achieving better results. This observation implies that an appropriate pre-training mask strategy could enhance a model's ability to capture molecular features.

**3D Coordinate Information Improves Molecular Latent Representation**   When comparing models utilizing 2D graph-only information (AttrMasking, GraphMAE, and Mole-BERT) with those integrating 3D coordinate information (3D Denoising, Frad, SliDe, Uni-Mol, Uni-Mol+), a consistent trend emerges: the 3D-oriented models consistently outperform 2D graph-only models. This consistent superiority suggests that 3D information significantly enriches the representation of small molecular features, leading to more robust and informative latent representations. This observation underscores the potential value of incorporating 3D structural data into molecular representation learning models, offering promising avenues for further advancements in the field.

**Fitting Force Fields Reveals Deeper Molecular Properties**   Further comparison of models that utilize 3D coordinate information reveals that physically informed pre-training models (3D Denoising, Frad, and SliDe) perform better than SE(3)-invariant Transformer-based models (Uni-Mol and Uni-Mol+) in capturing hydrogen bonds (*hbond*), electrostatic interactions (*ecoul*), and rotatable bond torsions (*erotb*). The 3D denoising, Frad, and SliDe models theoretically fit the force fields of small molecules. As reported in SliDe, the correlation coefficients of the estimated force fields are ranked as SliDe > Frad > 3D denoising, and our experimental results reflect the same trend. Additionally, in the *einternal* task, which is related to internal torsional energy, the SliDe method stands out among the other models. These results indicate that training strategies closely aligned with the intrinsic properties of small molecules, such as force fields, indeed enhance the model's ability to capture meaningful molecular representations.

**The Superiority of UniMAP**   We observe that UniMAP achieves the best performance among all deep-learning-based methods in Table 4. This outstanding excellence can be attributed to several key factors of UniMAP. Firstly, UniMAP's fragment-level masking and alignment strategies emphasize the functionality of molecular fragments, which are recognized as chemical semantic units and play a crucial role in determining the bioactivity of molecules. Furthermore, the integration of fingerprint regression and functional group prediction supervision enriches UniMAP's representation, allowing it to capture intrinsic molecular structural information, similar to descriptors, making it easier to fit chemical and physical tasks. Finally, the shared Transformer that fuses SMILES and molecular graphs in a single-stream approach may enhance the expressive power of molecular embeddings.

### 3.3   COMPARISON ANALYSIS OF BASELINE MODELS WITH FINE-TUNING TECHNIQUES

### 3.3.1   EXPERIMENTAL SETUP

For the fine-tuning setting, we select the best four models(UniMAP, Uni-Mol, Frad, SliDe) from the previous baselines and fine-tune them on five representative tasks covering all property categories. This approach is chosen due to the high cost of training all parameters on numerous regression tasks, as there are 9 properties for each of the 10 protein targets. We follow the same fine-tuning strategy as outlined in the original papers. We train a separate model for each protein target and simultaneously predict the five tasks. Additionally, each model is trained three times using different random seeds to ensure robust and consistent performance.

Table 4: **Linear Probe Results.** Average Pearson correlation coefficients of 10 protein targets. Cells are blue if the Pearson correlation coefficient is above 0.5 and yellow if it is below 0.5. The *Avg* column displays the average result across all 9 tasks.

| Model | Chemical | | Physical | | | | | Biological | | Avg |
|---|---|---|---|---|---|---|---|---|---|---|
| | lipo | hbond | evdw | ecoul | esite | erotb | einternal | docking | emodel | |
| AttrMasking | 0.540 | 0.341 | 0.529 | 0.381 | 0.328 | 0.618 | 0.286 | 0.453 | 0.512 | 0.443 |
| GraphMAE | 0.560 | 0.335 | 0.523 | 0.351 | 0.343 | 0.635 | 0.289 | 0.478 | 0.507 | 0.447 |
| Mole-BERT | 0.598 | 0.396 | 0.575 | 0.464 | 0.378 | 0.699 | 0.316 | 0.516 | 0.569 | 0.501 |
| Uni-Mol | 0.677 | 0.408 | 0.718 | 0.517 | 0.432 | 0.769 | 0.457 | 0.569 | 0.688 | 0.582 |
| Uni-Mol+$_{BASE}$ | 0.606 | 0.343 | 0.664 | 0.408 | 0.377 | 0.676 | 0.418 | 0.499 | 0.609 | 0.511 |
| Uni-Mol+$_{LARGE}$ | 0.604 | 0.346 | 0.661 | 0.412 | 0.381 | 0.663 | 0.405 | 0.495 | 0.607 | 0.508 |
| 3D Denoising | 0.605 | 0.401 | 0.599 | 0.452 | 0.367 | 0.796 | 0.379 | 0.526 | 0.588 | 0.524 |
| Frad | 0.621 | 0.393 | 0.620 | 0.451 | 0.380 | 0.821 | 0.396 | 0.535 | 0.605 | 0.536 |
| SliDe | 0.645 | 0.404 | 0.643 | 0.459 | 0.385 | 0.834 | 0.443 | 0.540 | 0.617 | 0.552 |
| UniMAP | 0.688 | 0.475 | 0.719 | 0.550 | 0.459 | 0.840 | 0.490 | 0.601 | 0.694 | 0.613 |
| Descriptors$_{2D}$ | 0.721 | 0.484 | 0.759 | 0.572 | 0.494 | 0.888 | 0.508 | 0.649 | 0.742 | 0.646 |
| Descriptors$_{3D}$ | 0.553 | 0.357 | 0.650 | 0.371 | 0.271 | 0.606 | 0.394 | 0.410 | 0.559 | 0.463 |
| Descriptors$_{2D\&3D}$ | 0.722 | 0.486 | 0.760 | 0.573 | 0.495 | 0.889 | 0.509 | 0.651 | 0.744 | 0.647 |

### 3.3.2 RESULTS & ANALYSIS

Table 5: **Fine-Tuning Results.** Average RMSE and Pearson correlation coefficients for 10 protein targets. The best result is shown in **bold**, and the second-best result is underlined.

| Model | hbond | | ecoul | | esite | | docking | | emodel | |
|---|---|---|---|---|---|---|---|---|---|---|
| | RMSE↓ | Pearson↑ | RMSE↓ | Pearson↑ | RMSE↓ | Pearson↑ | RMSE↓ | Pearson↑ | RMSE↓ | Pearson↑ |
| Uni-Mol | **0.146** | **0.557** | **2.274** | **0.642** | **0.051** | **0.574** | **0.730** | 0.741 | 6.528 | 0.811 |
| UniMAP | 0.149 | 0.550 | 2.307 | 0.640 | 0.052 | 0.556 | 0.741 | **0.745** | 6.727 | **0.812** |
| Frad | 0.161 | 0.408 | 2.381 | 0.592 | 0.059 | 0.350 | 0.822 | 0.652 | 6.818 | 0.790 |
| SliDe | 0.156 | 0.507 | 2.487 | 0.577 | 0.056 | 0.490 | 0.763 | 0.710 | **6.466** | 0.810 |

In Table 5, we show the average RMSE and Pearson correlation coefficient for 10 protein targets. The results for each protein target with three random seeds are listed in Appendix 5.7. Notably, we can observe that the performance of all models improved significantly after fine-tuning compared to their linear probe results, surpassing the performance of previous descriptors. This indicates that pre-training models are better suited for the fine-tuning setting to enhance the performance of deep latent features. Furthermore, Table 12 demonstrates that the standard deviations across results under various seeds are negligible, indicating remarkable stability in MoleculeCLA's testing results. When comparing various models, we observe that Uni-Mol and UniMAP outperform SliDe and Frad, with SliDe performing better than Frad. This alignment in performance ranking with previous evaluation settings highlights the robustness and consistency of MoleculeCLA.

### 3.4 DRUG TARGET INTERACTION TASK

Though MoleculeCLA is curated through a computational approach, it can still effectively indicate model performance in real scenarios using experimental data as labels. In this section, we use the ligand binding affinity task, which has a closer correlation with the properties in MoleculeCLA, to verify this point.

### 3.4.1 EXPERIMENTAL SETUP

We integrate the encoders of different MRL methods into a versatile framework BindNet Feng et al. (2023a). This framework uses existing pre-trained encoders to provide molecular ligand representations, which are then combined with protein representations generated by a pre-trained

pocket encoder. These combined representations pass through a shared Transformer to predict affinity. We use complex data with binding affinity labels sourced from PDBBind Wang et al. (2005). Following Atom3D Townshend et al. (2020), this dataset has two different split settings: LBA 30% and LBA 60%, where the number indicates the protein sequence identity threshold. The Pearson correlation coefficient, Root Mean Square Error (RMSE), and Spearman correlation coefficient are the metrics used to evaluate different MRL encoders.

### 3.4.2 RESULTS & ANALYSIS

The performance of different MRL methods on the LBA task is shown in Table 6. We observe a positive correlation between the performance of these methods and their performance in MoleculeCLA as shown in Table 4: models that perform better in MoleculeCLA tend to achieve superior results in affinity prediction. These results demonstrate that, despite being derived from a computational approach, MoleculeCLA can effectively reflect the ability of molecular representation learning in experimental domain tasks. This indicates that our dataset has a broader and more general impact.

Table 6: Performance comparison of different MRL methods on LBA 30% and LBA 60% datasets. The best result is shown in **bold**, and the second-best result is underlined.

| Methods | LBA 30% | | | LBA 60% | | |
|---|---|---|---|---|---|---|
| | RMSE↓ | Pearson↑ | Spearman↑ | RMSE↓ | Pearson↑ | Spearman↑ |
| AttrMasking | 1.54 | 0.549 | 0.523 | 1.33 | 0.768 | 0.763 |
| GraphMAE | 1.50 | 0.548 | 0.537 | 1.29 | 0.772 | 0.765 |
| Mole-BERT | 1.44 | 0.572 | 0.567 | 1.29 | 0.777 | 0.777 |
| Uni-Mol | **1.34** | **0.632** | **0.622** | 1.23 | 0.793 | 0.788 |
| UniMAP | 1.38 | 0.617 | 0.612 | 1.24 | **0.797** | **0.797** |

## 4 CONCLUSION

In this work, we present MoleculeCLA: a large-scale dataset consisting of approximately 140,000 small molecules derived from computational ligand-target binding analysis, providing 9 properties that cover chemical, physical, and biological aspects. Our experiments demonstrate the importance of constructing a large-scale, scalable dataset, enabling more reliable and noise-free evaluation of molecular representation models. We introduce a novel methodology that does not directly rely on wet experimental data. Instead, our approach uses ligand-target binding analysis to establish a connection between real-world properties and computational metrics. This method avoids the inherent noise of wet experiments, eliminates batch effects, and ensures consistent properties for the same molecules. We believe our work not only provides a reliable benchmark that contributes to advances in the development of molecular representation learning but also serves as a successful case study in computational approaches, potentially inspiring benchmark construction in other domains within AI for Science.

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

# 5 APPENDIX

## 5.1 DATA AND CODE ACCESS

We make our data and evaluation code publicly available for research, ensuring that all results are easily reproducible. The specific access URLs are as follows:

- Download dataset: `https://huggingface.co/datasets/anonymousxxx/MoleculeCLA`

- Evaluation code and dataset usage document (including the code to load data, linear probe, etc.): `https://anonymous.4open.science/r/MoleculeCLA-96F0`

## 5.2 RELATED WORK

### 5.2.1 MOLECULAR REPRESENTATION LEARNING

Various molecular pre-training methods have been proposed to leverage large amounts of unlabeled data to learn meaningful molecular representations, benefiting different downstream tasks. There are three common pre-training strategies: Masking, which involves masking atoms or substructures in the 1D SMILES Weininger (1988) or 2D graphs of molecules and training the model to recover the masked content Hu et al. (2019); Hou et al. (2022); Xia et al. (2022); Feng et al. (2023c) ; Contrastive Learning, which aligns representations of different molecular modalities for modality fusion Liu et al. (2021); Feng et al. (2023c); Stärk et al. (2022) ; and 3D Denoising, a physically informed pre-training strategy that adds noise to the coordinates of 3D molecular conformers and trains the model to predict the noise, aiming to approximate learning the molecular force field Zaidi et al. (2022); Feng et al. (2023b); Ni et al. (2023), has demonstrated its superior performance on a variety of physical quantum tasks.

### 5.2.2 MOLECULAR BENCHMARKS

Existing methods typically evaluate their performance on molecular property prediction tasks. MoleculeNet Wu et al. (2018) is a commonly used benchmark that includes multiple subtasks covering molecular physicochemical and biological properties, with labels organized into classification and regression tasks. QM9 Ruddigkeit et al. (2012); Ramakrishnan et al. (2014) and MD17 Chmiela et al. (2017) are quantum property-related tasks that include energetic and electronic properties and molecular force field predictions as regression tasks. Since quantum properties are highly related to 3D conformers, these tasks are often used as benchmarks for 3D-based methods. Moreover, drug target interaction datasets such as KIBA Tang et al. (2014), Davis Davis et al. (2011), and LBA Townshend et al. (2020) are evaluated for binding affinity prediction tasks, integrating molecular pre-trained encoders to extract molecular ligand representations. Additionally, several benchmarks Zhang et al. (2023); van Tilborg et al. (2022) concentrate on bioactivity cliffs, wherein structurally similar molecular pairs exhibit significant disparities in potency.

### 5.2.3 COMPARISON WITH OTHER COMPUTATIONAL DATASETS

There are also computational datasets in other domains. The Harvard Clean Energy Project Hachmann et al. (2011) features an automated, high-throughput framework for evaluating millions of molecular motifs via quantum chemistry. Lyu et al. (2019) explores structure-based docking of a library comprising 170 million compounds from 130 reactions, aiming to create an extensive library for identifying novel chemotypes. DOCKSTRING García-Ortegón et al. (2022) constructs a dataset that includes docking scores and poses and develops a series of pharmaceutically relevant benchmark tasks, such as virtual screening and de novo design of selective kinase inhibitors. Different from previous computational approaches, MoleculeCLA is specifically designed as a benchmark for molecular representation learning methods. Beyond evaluating final docking scores, MoleculeCLA incorporates intermediate, clearly defined components relevant to various molecular properties as regression tasks for assessing different molecular representation learning methods.

Table 7: The mean and standard deviation values of samples across the train, validation, and test datasets under scaffold split

| Property | Train Dataset | | Validation Dataset | | Test Dataset | |
|---|---|---|---|---|---|---|
| | Mean | Std | Mean | Std | Mean | Std |
| lipo | -2.78 | 0.72 | -2.83 | 0.71 | -2.84 | 0.72 |
| hbond | -0.17 | 0.18 | -0.16 | 0.18 | -0.16 | 0.18 |
| evdw | -37.18 | 6.52 | -38.14 | 6.41 | -38.17 | 6.44 |
| ecoul | -4.51 | 3.12 | -4.44 | 3.00 | -4.44 | 3.01 |
| esite | -0.06 | 0.06 | -0.06 | 0.06 | -0.06 | 0.06 |
| erotb | 0.70 | 0.31 | 0.70 | 0.28 | 0.70 | 0.27 |
| einternal | 5.44 | 4.87 | 5.50 | 3.84 | 5.48 | 3.82 |
| docking | -6.43 | 1.12 | -6.49 | 1.11 | -6.49 | 1.11 |
| emodel | -55.54 | 11.35 | -56.70 | 11.23 | -56.79 | 11.23 |

## 5.3 DATASET DETAILS

We chose 10 representative protein targets covering multiple categories to capture the comprehensive characteristics of small molecules. The categories include Kinase, G-Protein Coupled Receptor, Ion Channel, Nuclear Receptor, Cytochrome, Epigenetic, and Viral proteins. Detailed information on all protein targets can be found in the Table 2. Each protein target will have 9 molecular property tasks, including chemical properties (*lipo*, *hbond*), physical properties (*evdw*, *ecoul*, *esite*, *erotb*, *einternal*), and biological properties (*docking*, *emodel*). Task details are listed in Table 1. The average and standard deviation values of samples in the train, validation, and test sets for various tasks under the scaffold split are shown in Table 7.

The docking process is detailed as follows: Initially, up to 32 stereoisomers were generated for each small molecule, with ionization states optimized for physiological pH. The protein structures were then prepared by adding hydrogen atoms, predicting ionization states, and removing water molecules. A docking grid, centered on the co-crystallized ligand, was defined with dimensions of 20 Å. Docking was performed using the default Glide SP settings. Only molecules that successfully docked to all ten targets were retained. The resulting docking poses and associated values were compiled to construct the final dataset.

## 5.4 BASELINE MODEL DESCRIPTION

**AttrMasking** Hu et al. (2019) In this model, input node and edge attributes (such as atom types in a molecular graph) are randomly masked. The Graph Neural Network (GNN) is then trained to predict the masked attributes.

**GraphMAE** Hou et al. (2022) GraphMAE pre-trains a GNN by masking input node features with a [MASK] token and encoding the corrupted graph. During decoding, it re-masks selected node codes with a [DMASK] token and uses a GNN decoder to reconstruct the masked node features, optimizing with scaled cosine error.

**Mole-BERT** Xia et al. (2022) Mole-BERT paper proposes a context-aware tokenizer, encoding atom attributes into meaningful discrete codes. With the expanded atom "vocabulary", the authors introduce a novel node-level pre-training task called masked atoms modeling and triplet masked contrastive learning for graph-level pre-training.

**Uni-Mol** Zhou et al. (2023) Uni-Mol is a universal 3D molecular representation learning framework designed to enhance the representation ability and application scope of MRL methods by incorporating 3D information. It features two pretrained models with SE(3)-Transformer architecture: one trained on 209M molecular conformations and another on 3M protein pocket data. Uni-Mol also includes fine-tuning strategies for various downstream tasks.

**Uni-Mol+** Lu et al. (2023) Uni-Mol+ addresses the challenge of accurately predicting quantum chemical (QC) properties by leveraging 3D equilibrium conformations. Unlike previous methods using 1D SMILES sequences or 2D molecular graphs, Uni-Mol+ starts with a raw 3D molecule

conformation generated by inexpensive methods like RDKit. This conformation is iteratively updated to its target DFT equilibrium conformation using neural networks.

**3D Denoising** Zaidi et al. (2022)  This paper presents a pre-training technique based on denoising for molecular property prediction from 3D structures, particularly addressing the challenge of limited data. By using large datasets of 3D molecular structures at equilibrium, the method learns meaningful representations for downstream tasks. The denoising objective, linked to score-matching, corresponds to learning a molecular force field from equilibrium structures.

**Frad** Feng et al. (2023b)  Frad introduces a hybrid noise strategy, applying noise to both dihedral angles and coordinates, and a fractional denoising approach that decouples these noises, focusing on coordinate denoising. This approach maintains force field equivalence and enhances sampling of low-energy structures.

**SliDe** Ni et al. (2023)  This paper introduces a new method for molecular pre-training called sliced denoising (SliDe), which is grounded in classical mechanical intramolecular potential theory. SliDe employs a novel noise strategy that perturbs bond lengths, angles, and torsion angles to improve the sampling of molecular conformations. It also uses a random slicing technique to avoid the computational burden of calculating the Jacobian matrix, which is crucial for estimating the force field.

**UniMAP** Feng et al. (2023c)  UniMAP is a universal SMILES-graph representation learning model designed to capture fine-grained semantics between SMILES and graph representations of molecules. It starts with an embedding layer to obtain token and node/edge representations, followed by a multi-layer Transformer for deep cross-modality fusion. UniMAP introduces four pre-training tasks: Multi-Level Cross-Modality Masking , SMILES-Graph Matching, Fragment-Level Alignment, and Domain Knowledge Learning.

## 5.5 RESULTS OF EXPERIMENTS WITH MULTI-LAYER PERCEPTRON(MLP)

Table 8 shows the performance of the MLP, which takes different molecular latent representations or descriptors as input. The MLP architecture consists of 2 layers and utilizes LeakyReLU as the activation function in the middle layer. Training of the multi-layer perceptron model will span 50 epochs with a batch size of 128 and an initial learning rate of 1e-4. To facilitate learning, a cosine decay learning rate scheduler is employed, with a warmup ratio set to 0.02 of the total training steps. For optimization, we employ L1 loss and Adam optimizer, configured with $\beta_1 = 0.9$ and $\beta_2 = 0.999$. Each experiment is conducted on an NVIDIA A100-PCIE-40GB GPU and takes approximately half an hour to converge. When comparing the linear regression results in Table 4 and the MLP results in Table 8, it is evident that the benefits of the deep learning model's latent representation are more pronounced in the more complex MLP architecture compared to the simpler linear regression model.

Table 8: **MLP Model Results**: Average Pearson correlation coefficients for ten protein targets. The best result is shown in bold, and the second-best result is underlined.

| Model | Chemical | | Physical | | | | | Biological | | Avg |
|---|---|---|---|---|---|---|---|---|---|---|
| | lipo | hbond | evdw | ecoul | esite | erotb | einternal | docking | emodel | |
| AttrMasking | 0.483 | 0.239 | 0.497 | 0.318 | 0.223 | 0.542 | 0.268 | 0.394 | 0.467 | 0.376 |
| GraphMAE | 0.533 | 0.274 | 0.531 | 0.332 | 0.260 | 0.611 | 0.301 | 0.445 | 0.496 | 0.396 |
| Mole-BERT | 0.666 | 0.438 | 0.676 | 0.527 | 0.418 | 0.793 | 0.429 | 0.612 | 0.681 | 0.536 |
| Uni-Mol | 0.691 | 0.402 | 0.728 | 0.513 | 0.403 | 0.818 | 0.467 | 0.591 | 0.698 | 0.590 |
| Uni-Mol+$_{BASE}$ | 0.640 | 0.370 | 0.695 | 0.456 | 0.384 | 0.744 | 0.453 | 0.556 | 0.661 | 0.522 |
| Uni-Mol+$_{LARGE}$ | 0.627 | 0.358 | 0.683 | 0.445 | 0.371 | 0.721 | 0.438 | 0.537 | 0.645 | 0.510 |
| 3D Denoising | 0.647 | 0.410 | 0.667 | 0.495 | 0.359 | 0.838 | 0.436 | 0.588 | 0.662 | 0.530 |
| Frad | 0.653 | 0.403 | 0.671 | 0.487 | 0.368 | 0.855 | 0.443 | 0.592 | 0.660 | 0.533 |
| SliDe | 0.659 | 0.396 | 0.674 | 0.478 | 0.348 | 0.857 | 0.462 | 0.576 | 0.655 | 0.532 |
| UniMAP | **0.716** | **0.493** | **0.754** | **0.582** | **0.479** | **0.908** | **0.516** | **0.675** | **0.760** | **0.635** |
| Descriptors$_{2D}$ | 0.679 | 0.369 | 0.725 | 0.471 | 0.358 | 0.852 | 0.464 | 0.559 | 0.663 | 0.538 |

## 5.6 SUPPLEMENTARY RESULTS OF LINEAR PROBE EXPERIMENTS WITH LINEAR REGRESSION

To provide a comprehensive comparison with different models using various metrics, we present the average RMSE, MAE, and R2 results for 10 protein targets in Tables 9, 10, and 11 below. Each

experiment is conducted on a Linux server equipped with an AMD EPYC 7742 64-Core Processor CPU and takes approximately half an hour to complete.

Table 9: **Linear Probe RMSE Results**: Average RMSE for ten protein targets.

| Model | Chemical | | Physical | | | | | Biological | | Avg |
|---|---|---|---|---|---|---|---|---|---|---|
| | lipo | hbond | evdw | ecoul | esite | erotb | einternal | docking | emodel | |
| AttrMasking | 0.600 | 0.165 | 5.473 | 2.774 | 0.060 | 0.210 | 3.664 | 0.986 | 9.642 | 2.619 |
| GraphMAE | 0.591 | 0.166 | 5.502 | 2.824 | 0.060 | 0.209 | 3.662 | 0.975 | 9.708 | 2.633 |
| Mole-BERT | 0.569 | 0.162 | 5.263 | 2.646 | 0.059 | 0.191 | 3.626 | 0.943 | 9.181 | 2.515 |
| Uni-Mol | 0.502 | 0.159 | 4.428 | 2.579 | 0.057 | 0.167 | 3.374 | 0.904 | 8.191 | 2.262 |
| Uni-Mol+$_{BASE}$ | 0.554 | 0.165 | 4.776 | 2.738 | 0.059 | 0.197 | 3.474 | 0.956 | 8.858 | 2.420 |
| Uni-Mol+$_{LARGE}$ | 0.555 | 0.165 | 4.798 | 2.735 | 0.059 | 0.200 | 3.498 | 0.959 | 8.888 | 2.429 |
| 3D Denoising | 0.563 | 0.161 | 5.144 | 2.664 | 0.059 | 0.162 | 3.533 | 0.935 | 9.014 | 2.471 |
| Frad | 0.553 | 0.161 | 5.037 | 2.666 | 0.059 | 0.153 | 3.505 | 0.929 | 8.882 | 2.438 |
| SliDe | 0.534 | 0.160 | 4.919 | 2.650 | 0.058 | 0.148 | 3.416 | 0.926 | 8.806 | 2.402 |
| UniMAP | 0.502 | 0.155 | 4.433 | 2.479 | 0.056 | 0.145 | 3.322 | 0.877 | 8.013 | 2.220 |
| Descriptors$_{2D}$ | 0.475 | 0.154 | 4.137 | 2.432 | 0.055 | 0.122 | 3.279 | 0.833 | 7.439 | 2.103 |

Table 10: **Linear Probe MAE Results**: Average MAE for ten protein targets.

| Model | Chemical | | Physical | | | | | Biological | | Avg |
|---|---|---|---|---|---|---|---|---|---|---|
| | lipo | hbond | evdw | ecoul | esite | erotb | einternal | docking | emodel | |
| AttrMasking | 0.478 | 0.134 | 4.242 | 2.170 | 0.046 | 0.138 | 2.778 | 0.775 | 7.458 | 2.024 |
| GraphMAE | 0.470 | 0.134 | 4.270 | 2.208 | 0.046 | 0.139 | 2.773 | 0.764 | 7.516 | 2.036 |
| Mole-BERT | 0.452 | 0.131 | 4.074 | 2.066 | 0.044 | 0.123 | 2.742 | 0.736 | 7.091 | 1.940 |
| Uni-Mol | 0.397 | 0.129 | 3.313 | 2.016 | 0.105 | 2.554 | 0.043 | 0.701 | 6.241 | 1.722 |
| Uni-Mol+$_{BASE}$ | 0.440 | 0.134 | 3.620 | 2.144 | 0.045 | 0.130 | 2.639 | 0.746 | 6.779 | 1.853 |
| Uni-Mol+$_{LARGE}$ | 0.441 | 0.134 | 3.643 | 2.140 | 0.045 | 0.133 | 2.656 | 0.749 | 6.820 | 1.862 |
| 3D Denoising | 0.448 | 0.130 | 3.959 | 2.079 | 0.045 | 0.101 | 2.669 | 0.731 | 6.952 | 1.902 |
| Frad | 0.440 | 0.131 | 3.871 | 2.082 | 0.045 | 0.092 | 2.645 | 0.725 | 6.836 | 1.874 |
| SliDe | 0.425 | 0.130 | 3.757 | 2.069 | 0.044 | 0.086 | 2.569 | 0.721 | 6.766 | 1.841 |
| UniMAP | 0.398 | 0.123 | 3.322 | 1.931 | 0.042 | 0.087 | 2.495 | 0.679 | 6.099 | 1.686 |
| Descriptors$_{2D}$ | 0.374 | 0.122 | 3.079 | 1.893 | 0.041 | 0.065 | 2.462 | 0.643 | 5.601 | 1.587 |

Table 11: **Linear Probe R2 Results**: Average R2 for ten protein targets.

| Model | Chemical | | Physical | | | | | Biological | | Avg |
|---|---|---|---|---|---|---|---|---|---|---|
| | lipo | hbond | evdw | ecoul | esite | erotb | einternal | docking | emodel | |
| AttrMasking | 0.293 | 0.118 | 0.277 | 0.146 | 0.111 | 0.374 | 0.081 | 0.205 | 0.261 | 0.207 |
| GraphMAE | 0.314 | 0.112 | 0.270 | 0.114 | 0.121 | 0.382 | 0.081 | 0.226 | 0.253 | 0.208 |
| Mole-BERT | 0.359 | 0.158 | 0.330 | 0.218 | 0.149 | 0.482 | 0.099 | 0.268 | 0.326 | 0.266 |
| Uni-Mol | 0.480 | 0.179 | 0.522 | 0.255 | 0.200 | 0.603 | 0.215 | 0.325 | 0.465 | 0.360 |
| Uni-Mol+$_{BASE}$ | 0.377 | 0.119 | 0.445 | 0.167 | 0.147 | 0.452 | 0.167 | 0.250 | 0.377 | 0.278 |
| Uni-Mol+$_{LARGE}$ | 0.375 | 0.122 | 0.440 | 0.169 | 0.150 | 0.432 | 0.156 | 0.246 | 0.373 | 0.274 |
| 3D Denoising | 0.370 | 0.163 | 0.359 | 0.207 | 0.140 | 0.627 | 0.144 | 0.278 | 0.349 | 0.293 |
| Frad | 0.390 | 0.157 | 0.386 | 0.206 | 0.150 | 0.667 | 0.157 | 0.287 | 0.369 | 0.308 |
| SliDe | 0.424 | 0.167 | 0.415 | 0.214 | 0.154 | 0.690 | 0.197 | 0.293 | 0.383 | 0.326 |
| UniMAP | 0.483 | 0.228 | 0.522 | 0.306 | 0.218 | 0.701 | 0.240 | 0.364 | 0.487 | 0.394 |
| Descriptors$_{2D}$ | 0.530 | 0.237 | 0.580 | 0.332 | 0.250 | 0.788 | 0.258 | 0.423 | 0.555 | 0.439 |

## 5.7 SUPPLEMENTARY RESULTS OF FINE-TUNING EXPERIMENTS

MoleculeCLA can provide more stable evaluation results. In this section, we list the fine-tuning results for each protein target with three random seeds in Table 12. Each fine-tuning experiment is conducted on an NVIDIA A100-PCIE-40GB, with durations ranging from 1 hour to 1 day, depending on the complexity of the pre-training methods.

Table 12: **Performance of Different Models Fine-Tuning.** Pearson correlation coefficients for different protein targets.

| Target | Model | hbond | ecoul | esite | docking | emodel |
|---|---|---|---|---|---|---|
| ABL1 | Uni-Mol | 0.602 (0.007) | 0.670 (0.003) | 0.594 (0.003) | 0.800 (0.002) | 0.872 (0.002) |
| | Frad | 0.460 (0.002) | 0.604 (0.003) | 0.414 (0.007) | 0.718 (0.007) | 0.859 (0.002) |
| | SliDe | 0.541 (0.000) | 0.604 (0.000) | 0.503 (0.000) | 0.779 (0.000) | 0.873 (0.000) |
| | UniMAP | 0.598 (0.002) | 0.671 (0.001) | 0.588 (0.001) | 0.802 (0.001) | 0.874 (0.001) |
| ADRB2 | Uni-Mol | 0.536 (0.002) | 0.666 (0.001) | 0.519 (0.007) | 0.758 (0.003) | 0.834 (0.002) |
| | Frad | 0.405 (0.004) | 0.620 (0.003) | 0.270 (0.005) | 0.689 (0.003) | 0.822 (0.002) |
| | SliDe | 0.468 (0.000) | 0.575 (0.000) | 0.397 (0.000) | 0.725 (0.000) | 0.837 (0.000) |
| | UniMAP | 0.534 (0.005) | 0.664 (0.002) | 0.510 (0.003) | 0.763 (0.002) | 0.836 (0.002) |
| GluA2 | Uni-Mol | 0.508 (0.003) | 0.532 (0.001) | 0.496 (0.006) | 0.718 (0.003) | 0.810 (0.002) |
| | Frad | 0.389 (0.016) | 0.489 (0.007) | 0.345 (0.005) | 0.638 (0.005) | 0.799 (0.005) |
| | SliDe | 0.461 (0.000) | 0.449 (0.000) | 0.388 (0.000) | 0.680 (0.000) | 0.809 (0.000) |
| | UniMAP | 0.499 (0.002) | 0.530 (0.001) | 0.481 (0.002) | 0.718 (0.000) | 0.817 (0.000) |
| PPARG | Uni-Mol | 0.440 (0.003) | 0.497 (0.001) | 0.399 (0.005) | 0.762 (0.001) | 0.767 (0.002) |
| | Frad | 0.277 (0.005) | 0.422 (0.004) | 0.140 (0.023) | 0.677 (0.001) | 0.755 (0.003) |
| | SliDe | 0.373 (0.000) | 0.377 (0.000) | 0.284 (0.000) | 0.731 (0.000) | 0.778 (0.000) |
| | UniMAP | 0.423 (0.001) | 0.486 (0.003) | 0.325 (0.005) | 0.763 (0.001) | 0.771 (0.001) |
| CYT2C9 | Uni-Mol | 0.584 (0.001) | 0.603 (0.001) | 0.505 (0.003) | 0.696 (0.004) | 0.674 (0.005) |
| | Frad | 0.391 (0.006) | 0.517 (0.004) | 0.231 (0.019) | 0.602 (0.012) | 0.631 (0.005) |
| | SliDe | 0.493 (0.000) | 0.530 (0.000) | 0.402 (0.000) | 0.657 (0.000) | 0.660 (0.000) |
| | UniMAP | 0.568 (0.002) | 0.598 (0.001) | 0.483 (0.002) | 0.701 (0.003) | 0.676 (0.002) |
| HDAC2 | Uni-Mol | 0.684 (0.001) | 0.767 (0.001) | 0.759 (0.003) | 0.852 (0.001) | 0.892 (0.002) |
| | Frad | 0.505 (0.012) | 0.719 (0.003) | 0.535 (0.003) | 0.796 (0.005) | 0.873 (0.001) |
| | SliDe | 0.659 (0.000) | 0.738 (0.000) | 0.722 (0.000) | 0.846 (0.000) | 0.898 (0.000) |
| | UniMAP | 0.676 (0.003) | 0.766 (0.001) | 0.756 (0.000) | 0.854 (0.001) | 0.887 (0.000) |
| 3CL | Uni-Mol | 0.564 (0.003) | 0.691 (0.004) | 0.546 (0.006) | 0.683 (0.003) | 0.789 (0.002) |
| | Frad | 0.444 (0.011) | 0.659 (0.004) | 0.365 (0.002) | 0.580 (0.005) | 0.760 (0.001) |
| | SliDe | 0.520 (0.000) | 0.627 (0.000) | 0.459 (0.000) | 0.645 (0.000) | 0.783 (0.000) |
| | UniMAP | 0.564 (0.002) | 0.691 (0.002) | 0.528 (0.004) | 0.689 (0.002) | 0.789 (0.002) |
| HIVINT | Uni-Mol | 0.507 (0.004) | 0.614 (0.001) | 0.645 (0.003) | 0.677 (0.001) | 0.816 (0.002) |
| | Frad | 0.365 (0.010) | 0.587 (0.001) | 0.438 (0.009) | 0.584 (0.001) | 0.803 (0.001) |
| | SliDe | 0.465 (0.000) | 0.627 (0.000) | 0.604 (0.000) | 0.639 (0.000) | 0.811 (0.000) |
| | UniMAP | 0.503 (0.003) | 0.691 (0.002) | 0.528 (0.004) | 0.685 (0.003) | 0.819 (0.001) |
| KRAS | Uni-Mol | 0.629 (0.001) | 0.656 (0.003) | 0.586 (0.004) | 0.707 (0.002) | 0.809 (0.002) |
| | Frad | 0.505 (0.006) | 0.719 (0.003) | 0.360 (0.012) | 0.567 (0.008) | 0.784 (0.003) |
| | SliDe | 0.587 (0.000) | 0.593 (0.000) | 0.482 (0.000) | 0.664 (0.000) | 0.816 (0.000) |
| | UniMAP | 0.617 (0.003) | 0.649 (0.003) | 0.569 (0.001) | 0.706 (0.001) | 0.808 (0.001) |
| PDE5 | Uni-Mol | 0.518 (0.008) | 0.726 (0.000) | 0.688 (0.002) | 0.759 (0.003) | 0.841 (0.002) |
| | Frad | 0.336 (0.003) | 0.605 (0.003) | 0.396 (0.014) | 0.665 (0.009) | 0.821 (0.007) |
| | SliDe | 0.505 (0.000) | 0.593 (0.000) | 0.482 (0.000) | 0.730 (0.000) | 0.839 (0.000) |
| | UniMAP | 0.514 (0.000) | 0.687 (0.001) | 0.569 (0.001) | 0.771 (0.001) | 0.847 (0.001) |

## 5.8 AUTHOR STATEMENT, MAINTENANCE AND LICENSING

We acknowledge and accept full responsibility for any potential violation of rights and legal obligations arising from the use of the data provided in this paper. All the URLs are hosted on the stable website, ensuring all resources are available for a long time. Our dataset is available under the MIT

license. The evaluation code is hosted by the GitHub organization and uses the MIT license. We hope researchers will join this repository and further promote the research.

