# OpenReview forum: "MoleculeCLA: Rethinking Molecular Benchmark via Computational Ligand-Target Binding Analysis"
_ICLR.cc/2025/Conference — ICLR 2025 Conference Withdrawn Submission_

### Official Review · Reviewer_SiME · 2024-10-30

**Soundness:** 2
**Presentation:** 3
**Contribution:** 2
**Rating:** 8
**Confidence:** 2

**Summary:**

This paper introduces MoleculeCLA, a novel large-scale dataset to serve as a benchmark for evaluating molecular representation models. MoleculeCLA contains 10 protein targets and a wide and diverse range of molecular properties, including chemical, physical, and biological. The authors note that MoleculeNet, the current and commonly used benchmark, has multiple drawbacks, such as limited dataset size, label imbalances and inaccuracies, and general data inconsistencies, thus warranting a new and improved dataset. In the experiments, the authors used MoleculeCLA to evaluate nine representative deep learning models with varied architectures and representation approaches. The linear probe method is employed to assess the models’ abilities to capture abstract information within latent representations, using multiple regression metrics for evaluation. Additionally, some models were fine-tuned to predict five representative tasks, as well as the drug target interaction task. The authors were able to draw meaningful and insightful comparisons among the different baseline models.

**Strengths:**

- The authors present a good analysis of MoleculeCLA, showing it encompasses a diverse chemical space, characterized by a large number of protein targets and a wide range of molecular properties.

- Valid criticisms of MoleculeNet are proposed, underlying the motivations behind presenting a new benchmark. MoleculeCLA seems to address the limitations of MoleculeNet.

- The experiments and results seem to be sound. Nine different models are used, the datasets sizes are sufficient, and good insights are derived for the presented results, suggesting that this dataset can be successfully used for benchmarking.

**Weaknesses:**

- The authors state: “We introduce a novel methodology that does not directly rely on wet experimental data… This method avoids the inherent noise of wet experiments, eliminates batch effects, and ensures consistent properties for the same molecules. We believe our work not only provides a reliable benchmark…”
    - I understand the points that were made against wet experiments. However, a common criticism against computational predictors, especially in the case of binding affinity, is that they are inaccurate. The authors state that their work is reliable and avoids the inherent noise of wet experiments, however, if computational predictors do not necessarily reflect real-world outcomes, then this statement may be inaccurate. It be a valuable improvement if the authors demonstrate correlation between their results and actual wet-lab results on a subset of compounds.

- There is no mention of MoleculeCLA’s limitations. It would be beneficial for the authors to provide insights into potential improvements that could be incorporated in the next iteration of MoleculeCLA.


Minor edits:
- Line 58: add space between “data curation errors.” and “4) Inconsistency”
- Line 84: add space between “analysis” and “(Figure 2)”
- Line 89: MolecueCLA → MoleculeCLA
- Figure 2 caption: linear prob → linear probe

**Questions:**

- Why was Glide used over other docking software, such as Autodock Vina?

---

### Official Review · Reviewer_PHJf · 2024-11-01

**Soundness:** 1
**Presentation:** 3
**Contribution:** 1
**Rating:** 3
**Confidence:** 4

**Summary:**

The authors propose a dataset (and baselines) for molecular representation learning methods that is based on the software GLIDE.

**Strengths:**

- Efforts improve benchmarks for learning representations of molecules are highly important

**Weaknesses:**

A) Significance: The proposed dataset will not be useful for the community because it is based on several flawed assumptions.
- Ground truth provided by GLIDE: The authors calculate features with the docking software GLIDE which is used as a ground truth here. However, these are simulation-based values that can also be far from the actual ground truth. Machine learning methods can never get better than GLIDE, which is problematic because with sufficient data learning-based approaches should be able to perform better than simulation-based approaches.
- Neglecting characteristics of the field: humanity's collected knowledge about molecules comes from experiments. These experiments provide (usually) small data volumes, unbalanced label distributions, label noise, inconsistencies. These characteristics are in the nature of the field itself and should not be changed in order to keep an unbiased, and fair few on methods. However, the authors see these characteristics as unwanted, and change them. This changes their datasets and benchmark into a highly biased and unfair view on methods.


B) Lack of clarity: There is a severe lack of clarity when it comes to attributing prior work in the field. The first paragraph about molecular representation learning cites completely inappropriate papers that neglect pioneering work in the field and falsely attribute molecular representation learning. Learning representations of molecules with neural network goes back into the 1990, see eg reference [1], and Deep Learning methods started around 2014 (references [2-5]). Even graph neural networks for learning molecule representation originate from much earlier ([6]). The authors should completely re-work the paragraph on molecular representation learning.

Similarly, for the second paragraph on the datasets, there is a lack of refering to closely related work, to pioneering work, and the paragraph provides a highly biased view on the field. The paragraph, eg, misses the ExCAPE-db [7], LSC [8], and FS-Mol [9] datasets.

C) Severe technical errors: All reported performance metrics are done without nested cross-validation, proper hyperparameter selection, training reruns, and provided without error bars, confidence intervals and statistical test. Thus, all reported results could just arise by chance and conclusions drawn from those are likely wrong.

D) Originality: The dataset and the work done in this manuscript are new, but not relevant (see A)


References:
[1] Huuskonen, J., Salo, M., & Taskinen, J. (1998). Aqueous solubility prediction of drugs based on molecular topology and neural network modeling. Journal of chemical information and computer sciences, 38(3), 450-456.
[2] Unterthiner, T., Mayr, A., Klambauer, G., Steijaert, M., Wegner, J. K., Ceulemans, H., & Hochreiter, S. (2014, December). Deep learning as an opportunity in virtual screening. In Proceedings of the deep learning workshop at NIPS (Vol. 27, pp. 1-9). Cambridge, MA.
[3] Dahl, G. E., Jaitly, N., & Salakhutdinov, R. (2014). Multi-task neural networks for QSAR predictions. arXiv preprint arXiv:1406.1231.
[4] Lusci, A., Pollastri, G., & Baldi, P. (2013). Deep architectures and deep learning in chemoinformatics: the prediction of aqueous solubility for drug-like molecules. Journal of chemical information and modeling, 53(7), 1563-1575.
[5] Unterthiner, T., Mayr, A., Klambauer, G., & Hochreiter, S. (2015). Toxicity prediction using deep learning. arXiv preprint arXiv:1503.01445.
[6] Scarselli, F., Gori, M., Tsoi, A. C., Hagenbuchner, M., & Monfardini, G. (2008). The graph neural network model. IEEE transactions on neural networks, 20(1), 61-80.
[7] Sun, J., Jeliazkova, N., Chupakhin, V., Golib-Dzib, J. F., Engkvist, O., Carlsson, L., ... & Chen, H. (2017). ExCAPE-DB: an integrated large scale dataset facilitating Big Data analysis in chemogenomics. Journal of cheminformatics, 9, 1-9.
[8] Mayr, A., Klambauer, G., Unterthiner, T., Steijaert, M., Wegner, J. K., Ceulemans, H., ... & Hochreiter, S. (2018). Large-scale comparison of machine learning methods for drug target prediction on ChEMBL. Chemical science, 9(24), 5441-5451.
[9] Stanley, M., Bronskill, J. F., Maziarz, K., Misztela, H., Lanini, J., Segler, M., ... & Brockschmidt, M. (2021, August). Fs-mol: A few-shot learning dataset of molecules. In Thirty-fifth Conference on Neural Information Processing Systems Datasets and Benchmarks Track (Round 2).

**Questions:**

Can you elaborate on the dataset sizes and dataset characteristics of the drug discovery benchmarks (like LSC, ExCAPE-db, and FS-Mol) in relation to your proposed dataset?

---

### Official Review · Reviewer_JHdJ · 2024-11-04

**Soundness:** 2
**Presentation:** 3
**Contribution:** 2
**Rating:** 3
**Confidence:** 4

**Summary:**

This paper presents a new dataset, MoleculeCLA, for molecular property prediction, designed to address issues present in existing benchmarks, including label imbalance, limited data volume, and noisy labels. MoleculeCLA is generated through computational ligand-target binding analysis, providing a benchmark that includes chemical, physical, and biological properties.

**Strengths:**

1. The paper introduces a computational approach for generating molecular data, avoiding the noise and inconsistencies associated with experimental datasets, resulting in a scalable benchmark.
2. MoleculeCLA includes nine properties covering chemical, physical, and biological aspects, facilitating a comprehensive evaluation of molecular representation models.

**Weaknesses:**

1. The dataset is generated using computational methods, avoiding the noise of wet-lab experiments. However, Glide’s accuracy as a molecular docking method is not entirely convincing. The expectation for computational methods might be a level of accuracy similar to that of DFT in quantum chemistry.
2. Models trained on this dataset have not yet been tested in real-world tasks or demonstrated improvements on datasets with experimental labels. Similar to Section 3.4 in the paper, their effectiveness in other molecular property prediction tasks remains to be validated.
3. The dataset contains about 140,000 data points, but only 14,000 unique molecules. For data generated by computational methods, this scale is relatively samll.
4. Since the data is constructed using a docking method, molecular property predictions are limited to docking-related properties, resulting in restricted property coverage.

**Questions:**

1. As mentioned in point 2 under weaknesses, how do models trained on this dataset perform on experimental data? In other words, how can it be practically verified that this dataset will lead to improved model performance?
2. Is it possible to use this dataset for pretraining and then finetune on data with experimental labels?
3. What type of model is expected to be trained using MoleculeCLA? Is the goal to achieve Glide-level accuracy, and what tasks should the model be able to perform?
4. What are the future directions for expanding this benchmark? MoleculeCLA relies on computational ligand-target analysis; are there plans to include additional properties to broaden the benchmark's applicability?

---

### Official Review · Reviewer_B3xF · 2024-11-04

**Soundness:** 2
**Presentation:** 3
**Contribution:** 2
**Rating:** 3
**Confidence:** 3

**Summary:**

The paper introduces MoleculeCLA, a new computational benchmark dataset for molecular property prediction, specifically aimed at ligand-target binding analysis. This dataset, containing approximately 140,000 molecules, was developed to address limitations in current molecular benchmarks like MoleculeNet, which are often constrained by small datasets, label imbalance, and label noise. MoleculeCLA uses properties computed via Schrödinger's Glide software, covering chemical, physical, and biological aspects that are relevant to drug discovery. The authors demonstrate the dataset's utility by evaluating multiple molecular representation learning models and providing insights into the performance characteristics of different architectures.

**Strengths:**

- The benchmark presented in this paper targets a widely recognized issue within the field that requires further exploration. Proposing a high-quality dataset to validate the effectiveness of chemical pretrained models is essential.

- The authors provide a relatively comprehensive analysis and statistical description of the proposed dataset.

**Weaknesses:**

- The labels obtained through computational approaches, such as Glide, do not comprehensively reflect the application requirements, making it difficult to accurately assess the effectiveness of different molecular representation learning models. Glide is based on molecular mechanics empirical formulas, calibrated through experimental data. Unlike DFT, Glide and similar tools do not adhere to first-principles calculations when computing properties like docking scores. Although this dataset statistically benefits model training more than MoleculeNet, it demonstrate much limited diversity in task types compared to MoleculNet.

- There is an issue of fairness in the experimental evaluation. The proposed dataset serves as a downstream dataset, and while testing different pre-training methods, the authors downloaded pre-trained model parameters. However, the pre-training datasets used in these methods vary in scale and quality, such as UniMol, which combines ZINC and ChEMBL datasets (19M) as pretraining dataset. GraphMAE uses a subset of ZINC15 (2M) for pretraining. These differences in data quality and scale make comparisons of method effectiveness less convincing.

- The authors provide insufficient discussion on the choice of targets and properties. For instance, line 168 claims that *'the selection ensures broad coverage of potential interactions'*. Without supporting experiments or citations, it is difficult to verify this claim. Although Appendix 5.3 provides additional details, it still lacks a discussion on why these specific targets and tasks were chosen.

- Certain conclusions in the *Results & Analysis* section are questionable. For example, the conclusion on line 340 regarding the effectiveness of 3D coordinate information seems influenced by biases in the benchmark dataset, as molecular mechanics properties are closely tied to 3D information. However, this does not imply higher-quality molecular representations in a broader sense, such as for ADME/T tasks where 3D information may not be as critical. Similarly, the statements on line 359 regarding UniMAP’s superiority are not entirely objective, as UniMAP does not surpass UniMol in several experiments listed in the appendix.

- The selection of baseline methods in the benchmark is insufficient. Ignoring the impact of pre-training dataset quality and scale, the pre-training methods still lack coverage of many classic representation learning works, such as GEM [1], Molformer [2], KGPT [3], and KANO [4].

[1] Geometry-enhanced molecular representation learning for property prediction. Nat. Mach. Intell. 2022
[2] Geometry-enhanced molecular representation learning for property prediction.  Nat. Mach. Intell. 2023
[3] KPGT: Knowledge-Guided Pre-training of Graph Transformer for Molecular Property Prediction. Nat. Comm. 2023
[4] Knowledge graph-enhanced molecular contrastive learning with functional prompt. Nat. Mach. Intell. 2023

**Questions:**

Please refer to the statements in the Weaknesses.

**Details Of Ethics Concerns:**

The dataset proposed in this paper was obtained from commercially available library. Reorganizing and public these datasets may violate copyright issues.

---

### Official Review · Reviewer_aZBP · 2024-11-04

**Soundness:** 2
**Presentation:** 3
**Contribution:** 2
**Rating:** 3
**Confidence:** 4

**Summary:**

In this paper, the authors construct MoleculeCLA, a large-scale moleuclar representation dataset using the computational method Glide. MoleculeCLA containing about 140k molecules and 10 protein targets. Glide is employed to calcualte 9 properties for these molecules, covering chemical, physical, and biological properties. Based on MoleculeCLA, the authors evaluate the performance of various molecular representation learning methods and conduct analysis.

**Strengths:**

1. The authors proposed MoleculeCLA, a molecular representation dataset avoids the label noise, inconsistency, unbalanced label distribution, and data volume constraints problems existing in MoleculeNet.
2. Various molecular representation learning methods have been evaluated and analysised on the MoleculeCLA dataset, providing a deep understanding of these methods.
3. The paper is easy to follow and well-written.

**Weaknesses:**

1. The authors claim improved stability and reliability on MoleculeCLA (line 97), yet provide no comparative results on MoleculeNet. This omission prevents a robust assessment of the proposed model's performance against established benchmarks.
2. The authors utilize MoleculeCLA, a computationally generated dataset. However, unlike the QM9 dataset within MoleculeNet, which employs DFT-level calculations, the accuracy of the labels within MoleculeCLA is unclear and potentially unreliable. This raises concerns about the accuracy of labels and thus the evaluation result.
3. Unlike MoleculeNet, which offers wet-lab experimental labels for certain datasets, MoleculeCLA relies solely on computational predictions. Consequently, strong performance on MoleculeCLA merely indicates accurate prediction of GLIDE docking scores, not necessarily a correlation with real-world experimental outcomes. This gap limits the practical significance of the presented results.
4.  MoleculeCLA exhibits limitations compared to MoleculeNet, including lower protein diversity, and fewer tasks.

**Questions:**

1. Is the design of MoleculeNet takes the application in real-world scenarios into consideration? The practical scenarios inherently involve the label noise and class imbalance issues. Current MoleculeCLA can not reflect these problems.

---

### Note · Authors · 2024-11-13

I have read and agree with the venue's withdrawal policy on behalf of myself and my co-authors.